# Essential Amino Acids-Rich Diet Increases Cardiomyocytes Protection in Doxorubicin-Treated Mice

**DOI:** 10.3390/nu15102287

**Published:** 2023-05-12

**Authors:** Giovanni Corsetti, Claudia Romano, Evasio Pasini, Tiziano Scarabelli, Carol Chen-Scarabelli, Francesco S. Dioguardi

**Affiliations:** 1Division of Human Anatomy and Physiopathology, Department of Clinical and Experimental Sciences, University of Brescia, 25123 Brescia, Italy; cla300482@gmail.com; 2Italian Association of Functional Medicine, 20855 Lesmo (Milan), Italy; evpasini@gmail.com; 3Center for Heart and Vessel Preclinical Studies, St. John Hospital and Medical Center, Wayne State University, Detroit, MI 48236, USA; tscarabelli@hotmail.com; 4Division of Cardiology, Richmond Veterans Affairs Medical Center (VAMC), Richmond, VA 23249, USA; chenscarabelli@hotmail.com; 5Department of Internal Medicine, University of Cagliari, 09042 Monserrato (Cagliari), Italy; fsdioguardi@gmail.com

**Keywords:** heart, doxorubicin, essential amino acids, Klotho, necroptosis, mice

## Abstract

Background: Doxorubicin (Doxo) is a widely prescribed drug against many malignant cancers. Unfortunately, its utility is limited by its toxicity, in particular a progressive induction of congestive heart failure. Doxo acts primarily as a mitochondrial toxin, with consequent increased production of reactive oxygen species (ROS) and attendant oxidative stress, which drives cardiac dysfunction and cell death. A diet containing a special mixture of all essential amino acids (EAAs) has been shown to increase mitochondriogenesis, and reduce oxidative stress both in skeletal muscle and heart. So, we hypothesized that such a diet could play a favorable role in preventing Doxo-induced cardiomyocyte damage. Methods: Using transmission electron microscopy, we evaluated cells’ morphology and mitochondria parameters in adult mice. In addition, by immunohistochemistry, we evaluated the expression of pro-survival marker Klotho, as well as markers of necroptosis (RIP1/3), inflammation (TNFα, IL1, NFkB), and defense against oxidative stress (SOD1, glutathione peroxidase, citrate synthase). Results: Diets with excess essential amino acids (EAAs) increased the expression of Klotho and enhanced anti-oxidative and anti-inflammatory responses, thereby promoting cell survival. Conclusion: Our results further extend the current knowledge about the cardioprotective role of EAAs and provide a novel theoretical basis for their preemptive administration to cancer patients undergoing chemotherapy to alleviate the development and severity of Doxo-induced cardiomyopathy.

## 1. Introduction

Doxorubicin (Doxo, alias adriamycin) is a widely prescribed anthracycline drug that is very effective against many human malignant cancers. Unfortunately, the utility of anthracyclines is limited by their toxicity, in particular a progressive induction of congestive heart failure (CHF), which is dose- and time-dependent and usually refractory to common medications [1,2,3,4]. The estimated risk of CHF after the initiation of anthracycline chemotherapy increased to 2% at 2 years, and 5% at 15 years [5,6], respectively. The incidence of cardiomyopathy greatly increases with higher doses of anthracyclines and/or their combined use with other drugs. Although the mechanisms of action of anthracyclines in tumor cells remain a matter of controversy, the suggested mechanisms include: (1) inhibition of macromolecules synthesis by interference with DNA; (2) generation of reactive oxygen species (ROS) with lipid peroxidation; (3) DNA damages; (4) direct membrane effects; and (5) induction of apoptosis in response to topoisomerase-II inhibition [7,8]. Apoptosis is believed to be an important contributor to cardiotoxicity, causing loss of contractile units, ultimately leading to heart failure [9,10]. Doxo-induced cardiotoxicity and antitumor activity seems linked to very different mechanisms. The main damage induced by Doxo to the heart presents morphologically as dilation of the heart chambers and fibrosis, accompanied by vacuolization of cardiomyocytes, reduction of myofibrils, mitochondrial disruption, and chromatin disorganization [11]. Therefore, Doxo acts primarily as a mitochondrial toxin. Indeed, mitochondrial damage with consequent ROS increase and generation of oxidative stress, are central to Doxo-induced cardiac dysfunction and cell death [12,13,14]. Cardiomyocytes are very sensitive to ROS because they possess a lower content of anti-oxidative enzymes compared to mammalian cells in other organs [12]. Oxidative stress plays an important role in the pathophysiology of cardiovascular diseases. Excess in ROS causes generalized cellular dysfunction and protein and lipid peroxidation as well as DNA damage, promoting a large number of pathological conditions, such as cell damage and apoptotic cell death [15].

Recently, a humoral factor with important protective effects on cells, named Klotho, has attracted the attention of many researchers. Klotho is a protein showing anti-aging properties that reduces cellular oxidative stress, thus preventing cell death and apoptosis [16]. Two distinct forms have been currently identified: α-Klotho (referred to simply as Klotho), which is a 130-kDa protein found in the kidney, but also in others organs; and β-Klotho, a protein that shares 41% amino acid identity with α-Klotho, and is expressed in adipose tissue, the pancreas, and the liver [17]. Higher Klotho levels are associated with a lower cardiovascular risk, thereby suggesting a possible protective role for Klotho in cardiovascular diseases [18]. Klotho is also expressed in human cardiomyocytes and its expression is down-regulated in higher cardiovascular risk patients, while expression of stress-related molecules is increased [19].

In the failing human heart, the annual rate of cardiomyocyte cell loss is about 20–25%, predominantly oncotic necrosis and autophagic cell death (about 20%), whilst the remainder of the myocyte population dies from apoptosis [20]. Recently, it was demonstrated that autophagy precedes and promotes the occurrence of apoptosis, oncosis, and necroptosis, which only rarely start independently from autophagy [21]. As a consequence, mixed types of cell death may also occur, namely “autophagic-necrosis” [22,23] and/or “necroptosis”, which result from failure of the compensatory, pro-survival autophagic response [24].

Necroptosis is a well-characterized cell damaging process, combining morphological features of accidental cell death and maladaptive autophagy. Necroptosis involves loss of membrane integrity and release of damage-associated molecular pattern molecules (DAMPs), resulting in secondary inflammatory response [25]. Biochemically, necroptosis is a programmed process orchestrated by a complex set of proteins involving Receptor Interacting Protein-1 and -3 (RIP1 and RIP3), as well as mixed lineage kinase domain-like protein (MLKL). RIPs constitute a family of seven members, all of which contain a kinase domain that play a crucial role as regulators of cell survival and death [26]. Activation of the RIP1-RIP3-MLKL signaling pathway leads to disruption of cation homeostasis, plasma membrane rupture, and finally cell death [27]. RIP1, which is constitutively expressed in many tissues, represents a crucial adaptor kinase on the crossroad of stress-induced signaling pathways and a cell’s decision to live or die [28]. In addition, RIP1 is crucial for activating nuclear factor κB (NFκB) and its role has been found to extend to necrotic cell death [29]. TNFα treatment and T-cell activation can also induce RIP1 expression [28,30]. Furthermore, RIP3 is a critical regulator of programmed necrosis/necroptosis. Its activation is tightly regulated by a biochemical cascade that drives to the assembly of a macromolecular complex termed necrosome, and finally to cell death by necroptosis rather than apoptosis [31,32,33]. In addition, RIP3 can promote inflammation independent of its pro-necrotic activity [34]. The anti-necroptotic role of Klotho was demonstrated in the kidney, where Klotho protects tubular epithelial cells from ischemia-reperfusion injury, attenuating the elevation in RIP1 and RIP3, thereby inhibiting oxidative stress [35].

The majority of enzymes are proteins comprised of amino acids (AAs) linked together in one or more polypeptide chains. Therefore, activation, maintenance, or improvement of the synthesis of enzymes protecting from chemotherapy-linked stress is primarily dependent on the availability of AAs, and particularly of essential AAs (EAAs), from which all other non-EAAs may be metabolically derived.

Previous work has demonstrated that feeding mice a special diet rich in all EAAs in a stoichiometric ratio formulated according to human need, increased mice survival [36], and counteracted damage induced by senescence [37] and toxic molecules [38,39,40,41]. The positive effects of special EAA mixture supplementation were due to increased mitochondriogenesis, improved mitochondrial function, and reduced oxidative stress both in skeletal muscle and heart [42].

Because at present there is no specific treatment for Doxo cardiomyopathy, we hypothesize that a diet rich in free EAAs in a stoichiometric ratio according to human need could play a favorable role in preventing cardiomyocyte damage and/or death, by increasing Klotho expression and decreasing necroptotic markers. To this end, we evaluated by immunohistochemistry the expressions of Klotho, RIP1, and RIP3 in mice treated with Doxo and fed a special diet deprived of proteins but containing amounts of EAAs significantly higher than those derived from alimentary proteins. In addition, we assessed the ultrastructural morphology of cardiomyocytes along with the changes in enzymes that play a crucial role in inflammation and cell defense systems, such as TNFα, NFkB, IL1, SOD1, and citrate synthase.

## 2. Materials and Methods

### 2.1. Animals and Treatments

A total of 22 6-month-old male Balb/C mice (from Envigo, Holland) were individually housed in plastic cages with white wood chips for bedding, in a quiet room, under controlled lighting (12 h day/night cycle) and temperature (22 ± 1 °C) conditions. At all times, the animals had access to food and water that was supplied ad libitum. Every three days, body weight, diet, and water consumption were measured and restored. Animals were regularly evaluated by veterinary doctors for their health and maintenance of normal daily and nocturnal behavioral activities and for criteria of increased disease burden according to ethics standards for animal studies. The experimental protocol was approved by and conducted in accordance with the Italian Ministry of Health and complied with the ‘The National Animal Protection Guidelines’. The Ethics Committee for animal experiments of the University of Brescia (OPBA, Organismo Preposto per il Benessere Animale = Organism Controlling Animal Wellbeing) and the Italian Ministry of Health all approved the procedures.

The animals were divided into two groups. The first group (n = 11) was fed a standard laboratory diet (StD group). The second group (n = 11) was fed a special diet containing as an exclusive source of nitrogen a particular mixture of EAAs (EAAs group). The composition of the diets is presented in Table 1. From each group, 6 mice were ip injected with a single Doxo-HCl (15 mg/kg, from Sigma-Aldrich, St. Louis, MO, USA) (StD + Doxo and EAA + Doxo) to induce cardiomyopathy [43], while the remaining 5 animals served as the control group and were injected with 100 µL of saline solution. Two weeks later, the animals were euthanized by cervical dislocation. Their hearts were quickly removed and placed in an ice-cold saline solution. The samples dedicated to histochemistry (HC) and immunohistochemistry (IHC) were embedded in paraffin according to standard procedure. A small portion of the left ventricle was destined for ultrastructural examination.

### 2.2. Transmission Electron Microscopy (TEM)

Small samples of left ventricles were fixed in a solution composed of 2% paraformaldehyde plus 2% glutaraldehyde in 0.1 M cacodylate buffer (pH 7.4). Samples were then postfixed for 1 h in 1% buffered osmium tetroxide (OsO_4_). The pH was adjusted to 7.4 and the osmolarity to 330 milliosmoles per liter to assure dimensional stability of the specimens. Furthermore, after dehydration with sequential steps of increasing acetone concentration, the samples were processed with standard procedures for embedding in Araldite (Sigma Chemical Co, Milan, Italy). The thick sections (about 1 µm) were stained with 1% toluidine blue in 1% borax. Ultrathin sections (70 nm) were stretched with chloroform to eliminate compression and mounted on Formvar-filmed copper grids prior to staining with 2% aqueous uranyl acetate and lead citrate. Grids were examined with a Philips CM10 electron microscope and digital micrographs were captured. Data were collected from randomly selected areas of each sample at a final magnification of ×8900, examining several different levels of each sample. The mean total volume examined in each sample was about 6000 µm^3^.

### 2.3. Morphometry

All measurements were obtained using standard morphometric techniques, as previously described [44]. We calculated: the total surface of cytoplasm (Scyt), the surface of mitochondria (Smit), and the mitochondria number (Nmit). From these data we calculated the ratio between Smit and Scyt (Smit/Scyt), the number of mitochondria in 100 μm^2^ of cytoplasm, also called mitochondria density (Nmit/100 μm^2^), and the mean surface of mitochondria.

### 2.4. Histochemistry

Collagen production was evaluated with a picrosirius stain (Sirius-red), as previously described [45]. Briefly, the sections were de-paraffinized, moisturized in distilled water, and immersed in 1% phosphomolybdic acid (Sigma-Aldrich, St. Louis, MO, USA), then covered with 0.1% (*w*/*v*) Sirius-red F3B (C.I.35780 Science Lab, Huston, TX, USA) in saturated picric acid solution at room temperature. The sections were then washed in water and rapidly dehydrated, cleared in xylene, and mounted on a glass slide. The sections stained with Sirius-red were analyzed for collagen organization and fibrosis using a light microscope (Olympus BX50, Tokyo, Japan) under polarized light obtained with a polarizer filter (Olympus U-ANT, Tokyo, Japan). Under these conditions, collagen fibers of various thicknesses were stained differently. During tissue response to injury, fibronectin and type III collagen are synthesized in increased amounts, whereas in normal tissue the major constituent is type I collagen [46]. Although the birefringent color is more a measure of collagen fiber size than of collagen fiber type, usually the thicker and denser type I collagen fibers are detected as orange to red, whereas the thinner type III collagen fibers appear yellow to green [47,48].

### 2.5. Immunohistochemistry

For IHC, heart sections were incubated overnight with primary anti-RIPK1 (sc-133102), anti-Klotho (sc-22220), SOD1 (sc-11407), anti-GPx-4 (sc-50497), anti-IL-1β (sc-7884) from Santa Cruz Biotechnology Inc. (Dallas, TX, USA), anti-RIP3 (GTX 107574) from GeneTex, Inc. (Irvine, CA, USA), anti-TNFα (NB600-587) and anti-NFkB (NB110-57266) from Novus Biologicals (Centennial, CO, USA), and anti-Citrate Synthetase (ab96600) from Abcam, Inc. (Cambridge, UK). All polyclonal antibodies were diluted 1:100 with PBS; monoclonal antibodies were diluted 1:250 with PBS. The sections were processed according to the manufacturer’s protocol and visualized with a rabbit ABC-peroxidase staining system kit (Santa Cruz Biotechnology Inc., Dallas, TX, USA). Each set of experiments was performed in triplicate, with each replicate carried out under the same experimental conditions. The IHC control was performed by omitting the primary antibody in the presence of isotype-matched IgGs. The staining intensity in both HC and IHC slides was evaluated using an optical Olympus BX50 microscope equipped with an image analysis program (Image Pro Plus 4.5.1, Immagini e Computer, Milano, Italy) and analyzed quantitatively. The IOD (Integrated Optical Density) was calculated for arbitrary areas, by measuring 10 fields for each sample using a 20× lens.

On sections stained with anti-NFκB, we counted the percentage of immune-stained nuclei of cardiomyocytes. The number of anti-NFkB-stained nuclei present in 1 mm^2^ was determined on randomly chosen areas at 200× magnification, analyzing over 30 fields for each group. From these data and the area of the tissue section, positive nuclei were expressed as a % of the total number.

### 2.6. Statistics

Data are expressed as mean ± SD. Statistical analysis was performed by ANOVA followed by Bonferroni *t*-test (www.meta-calculator.com, accessed on 14 January 2023), to compare the results of four experimental groups. A value of *p* < 0.01 was considered statistically significant.

## 3. Results

### 3.1. Morphology

StD-fed mice that received Doxo did not show a change in body weight (bw), but exhibited a trend to lower heart weight (hw) and hw/bw ratio. Conversely, mice treated with Doxo and fed an EAAs diet did not display a change in either bw and/or hw (Table 2).

### 3.2. Histochemistry

Sirius-red staining for collagen in the heart of EAAs-fed animals showed the presence of finer collagen fibers than in StD-fed animals (Figure 1A,B). Doxo administration in animals fed with StD increased the degree of fibrosis, whereas in those fed with EAAs the degree of fibrosis did not increase, and the presence of collagen fibers appeared unchanged (Figure 1C,D).

### 3.3. Electron Microscopy

Cardiomyocytes from mice fed with StD and treated with Doxo frequently showed nuclei with irregular shapes, with introflexions of the nuclear membrane resembling bits. Sometimes, these nuclei appeared fragmented, with condensed chromatin and/or small clusters of chromatins near the nuclear envelope (Figure 2). Furthermore, we commonly observed impaired distribution and morphology of mitochondria that appeared irregular in shape and size (Figure 2A–D). The surface of mitochondria/total surface of cytoplasm (Smit/Scyt) ratio and the mitochondria density (Nmit/100 μm^2^) decreased compared to the mice fed with StD. The EAA feeding of Doxo-treated animals restored, at least partially, the morphometric parameters. Nuclear morphologies appeared unchanged from those of the control mice, without bite signs and/or fragmentation, although chromatin condensation was mildly increased (Figure 2E–G). The mitochondria morphology (Figure 2F–H) improved almost as much in the mice fed the EAAs diet alone. In both Doxo-treated groups, the mean size of mitochondria did not differ from that of the control group, although we noted a tendency to decrease in StD-fed mice. The myofibrils did not show signs of ultrastructural disorganization.

Morphometric data from mitochondria are summarized in Table 3.

### 3.4. Immunohistochemistry

Klotho staining intensity did not change before and after Doxo treatment in either diet (Figure 3A–E). However, compared to StD-fed mice, those fed the EAA diet showed more intense and homogeneous cell staining before and after Doxo administration (Figure 3D,E).

RIP1 staining increased significantly only in those animals that were StD-fed and treated with Doxo (Figure 4A–C). On the opposite side, EAA-fed mice did not exhibit increased staining after Doxo administration (Figure 4A,D,E).

Similarly, RIP3 immunostaining increased in StD-fed Doxo-treated mice (Figure 5A–C), whereas those fed with EAAs exhibited a tendency to reduced RIP3 intensity, which did not change after Doxo treatment (Figure 5A,D,E).

NFkB-stained nuclei (nr/100 μm^2^) increased strongly in StD-fed animals after Doxo administration (Figure 6A–C). The EAA diet led to an early increase of the nuclear staining for NFkB, which remained constant even after the administration of the chemotherapy (Figure 6A,D,E).

Cardiomyocytes from StD-fed mice treated with Doxo exhibited a greater TNFα staining intensity (Figure 7A–C). On the opposite side, no change was observed between mice fed EAAs alone and those fed EAAs plus Doxo treatment (Figure 7A,D,E).

IL-1β immunoreaction increased in both diets after chemotherapy (Figure 8A–C). However, the intensity of the immunostaining for IL-1β was greater in StD-fed mice, as compared to those fed EAAs (Figure 8A,D,E).

SOD1 staining increased intensely in StD-fed mice after Doxo administration (Figure 9A–C). On the contrary, EAA-rich food promoted high SOD1 expression both at baseline level and after Doxo administration (Figure 9A,D,E).

Glutathione peroxidase IV (GluPx-IV) immunoreaction increased in StD-fed mice after Doxo administration (Figure 10A–C). Mice fed a special diet containing EAAs alone increased GluPx-IV expression that increased moderately after Doxo treatment (Figure 10A,D,E).

Citrate synthase staining is located on the mitochondria and was therefore expressed as density (nr/100 μm^2^) of immune-stained dots. The intensely immunostained dots were numerous in both the StD and the EAA diets. (Figure 11A–C). After Doxo administration, we observed a strong decrease in number and stained intensity dots in StD-fed mice, whereas those EAA-fed showed limited decreases in both dot number and stained intensity (Figure 11A,D,E).

## 4. Discussion

Our data show for the first time that EAAs provided in large amounts by diet, so as to significantly increase EAA to NEAA ratios to >>1, can alleviate cardiac damage induced by chemotherapy by multiple mechanisms, i.e., improving mitochondria number density, augmenting Klotho and antioxidant enzyme levels, as well as reducing fibrosis, necroptotic, and inflammatory markers.

Previous works wherein rodents were fed a diet containing an excess of all EAAs in a stoichiometric ratio have shown an increase in mitochondriogenesis, improved mitochondria function, and reduced oxidative stress in skeletal muscle and heart [42]. In addition, in a pilot study, we demonstrated that supplementation of EAAs to Doxo-treated animals preserved cardiomyocytes architecture and improved the efficacy of chemotherapy [14,49]. More recently, we showed that a diet rich in EAAs favorably modified cell metabolism in different organs and experimental conditions [50,51], promoting survival [36]. Present data confirm and extend previous observations, highlighting that a diet rich in EAAs induced myocyte overexpression of Klotho, which acts as an important factor for cell survival by counteracting the synthesis of pro-necroptotic and pro-inflammatory markers.

The anti-aging protein Klotho is a humoral factor that has been found to protect cells from inflammation and oxidative stress, thus preventing cell death by apoptosis [16]. In cardiomyocytes exposed to hyperglycemia, it has been reported that Klotho plays a protective role both in vitro and in vivo, through the inactivation of ROS and NFκB-mediated inflammation [52]. In addition, in the kidney, the inhibition of oxidative stress induced by Klotho protected tubular epithelial cells from renal ischemic-reperfusion injury blocking necroptosis [35]. Previous work in rats treated with Doxo demonstrated that exogenous administration of Klotho attenuated Doxo-induced oxidative stress and apoptosis in cardiomyocytes, exerting a cardiac protective role by recovering the activation of the MAPKs/Nrf2 pathway [53]. Data from the present work confirm the role of Klotho in attenuating cardiomyocytes damage induced by chemotherapy. In addition, our data showed that a dietary excess of all EAAs led to Klotho overexpression in cardiac cells, protecting them from Doxo-induced damage and death.

Recently, a novel mechanism of cell death called programmed necrosis (or necroptosis), implicated in the pathogenesis of pathological conditions affecting different organs and apparatuses, has been identified. Necroptosis is a caspase-independent RIP3-mediated form of cell death [54]. The initiation of necroptosis requires several different stimuli, as well as the activity of RIP1 and RIP3 [55]. So, activation of RIP3 is regulated by the kinase RIP1 [56], which play a fundamental role in the modulation of cell fate in response to different stimuli [33,57].

In mice hearts, RIP3 overexpression can induce myocardial infarction, while the formation of a molecular complex among RIP1-RIP3 can drive cardiomyocytes to necrosis [58]. In human failing hearts, RIP3 expression was strongly enhanced in cardiac cells. Increased expression of RIP3 and RIP1 in human end-stage heart failure indicates the occurrence of both necroptosis activation and execution. Since RIP3/RIP1 overexpression was seen to frequently occur in LC3 positive myocytes, and only occasionally in those LC3 negative, it has been postulated that necroptosis was chiefly triggered [21].

Our data show that cardiomyocytes from mice fed StD and treated with Doxo overexpress RIP1 and RIP3, suggesting a concurrent activation of the necroptotic pathway. Indeed, studies clearly linked artificial overexpression of RIP1 to cell death, in both apoptosis and necroptosis [59]. However, when RIP1 expression is suppressed by gene knockout in MEFs or knockdown in cancer cells, DNA damage-induced cytotoxicity is significantly increased [60]. This suggests the importance of RIP1 modulation, and that cell survival signaling is predominant in RIP1-mediated genotoxic stress signaling, thus cell death pathways independent of RIP1 are sufficient to kill the cells. On the opposite side, an EAA-rich diet, although it shows a tendency to slightly increase RIP1 compared to StD, maintains lower levels of RIP3 even after treatment with chemotherapy, thus suggesting a preventive action on necroptosis. Indeed, many studies showed that RIP1 plays a key role in regulating cell death or survival, in response to different stimuli [26,29,61,62]. In addition, EAAs prevented the increase of TNFα in the Doxo-treated mice, while increasing the expression of NFkB, which seems to suggest a possible antiapoptotic activity. Although RIP1 is known to mediate cell death in response to DNA damage, it is also crucial for cell survival through activation of the NFkB pathway [63]. The anti-apoptotic activity of NFkB may avoid unnecessary and/or redundant cell death in response to apoptotic stimuli such as TNFα activation [64]. The RIP1-TNFR1-mediated NFκB activation is generally believed to be favorable for survival counteracting apoptosis [59]. Although NF-kB is a well-known transcriptional activator, it may function as a transcription repressor in certain circumstances, as when cells are responding to DNA damage [65]. It is likely that the activation of NFκB target genes and the cellular outcome in response to DNA damage-induced NFκB activation are dependent on the cellular context [66]. Accordingly, our findings documented that viable cardiomyocytes exhibited increased expression of RIP1 and nuclear NFkB, though not RIP3 expression, suggesting that the RIP1/NFkB pathway serves as an efficient survival pathway.

It has been demonstrated, both in vitro and in vivo, that ROS and NFκB-mediated inflammation were reduced by Klotho by augmenting nuclear factor erythroid 2-related factor 2 (Nrf2) expression [52]. Klotho deficiency has been shown to increase endogenous ROS generation and oxidative stress [67,68], while Klotho administration reduced oxidative stress and improved mitochondrial function [16,69,70,71]. In addition, Klotho-induced inhibition of insulin/IGF-1 signaling is associated with increased resistance to oxidative stress, which potentially contributes to its anti-aging properties [16].

It has recently been shown that dietary nutrition with a special mixture of EAAs averted Doxo-dependent mitochondrial damage and oxidative stress by KLF15/eNOS/mTORC1 signaling [72]. Accordingly, our results confirm the protective role of dietary nutrition with EAAs against chemotherapy-induced cardiac damage. The present data corroborate previous reports showing that animals treated with Doxo while receiving a special diet with EAAs exhibited enhanced synthesis of markers and modulators of cell survival, including Klotho, IL-1, SOD1, glutathione peroxidase, and citrate synthase. A schematic representation of the effects of the diet containing the special mixture of all EAAs in the stoichiometric ratio is summarized in the Figure 12.

### 4.1. Clinical Implications

We believe that the data obtained in this study have important clinical impact. Indeed, they show that a specific mixture of EAAs is able to reduce the cardiotoxic effects of Doxo, which greatly limits its clinical utilization. These data, barring clinical confirmation in man, will enable a broader use of Doxo even at an optimal dosage. In addition, it is known that this specific EAA mixture stimulates the synthesis of both muscle (counteracting sarcopenia) and globular (i.e., albumin) proteins, improving both quality of life and prognoses of cancer patients.

### 4.2. Limits of the Study

In this work, we predominantly focused on the morphological and enzymatic changes occurring in the cardiomyocytes of mice fed either an StD- or an EAA-rich diet while receiving Doxo administration. To this end, we relied almost exclusively on IHC, whose limitations represent the major weakness of the current work. Although IHC is not suited for accurate quantitative analysis, it offers several advantages. First, standard fixation maintains tissue architecture, allowing for meaningful morphological assessments. Second, IHC allows for the localization (both cellular and subcellular) of specific molecules, as well as for the quantification of their relative distributions between different cells types. However, the soundness of IHC is operator-dependent, requiring rigor of execution (to prevent technical errors potentially leading to misinterpretation) and analytical skills (to minimize the risk of bias). The longstanding experience of the operators in the specific field of IHC should vouch for the scientific accuracy of the data presented in our work. Nevertheless, we believe that the results emerging from this study are worthy of consideration and could represent a basis for further investigations aimed at improving the quality of life of patients by preventing damage deriving from chemotherapy.

## 5. Conclusions

Nutrition with a special dietary mixture of EAAs represents an important stimulus to support and maintain cellular metabolism, by activating cellular defense systems. This evidence reinforces the need for the early detection of chemotherapy-induced CHF and for the evaluation of effective cardio-protective strategies based on biological effectiveness. In our opinions, the results from this study improve our knowledge concerning the protective effects of a special mixture of EAA administration against pharmacological toxicity of anthracyclines, providing a novel theoretical basis and a rationale for the application of EAAs in alleviating chemotherapy-related cardiotoxicity in clinical practice.

## Figures and Tables

**Figure 1 nutrients-15-02287-f001:**
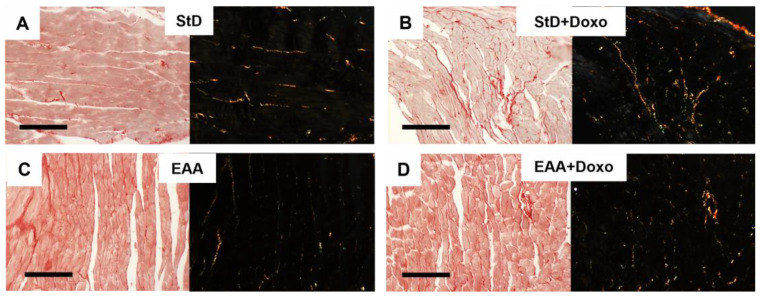
(**A**–**C**) Representative Sirius-red staining for collagen before and after Doxo administration according to diets under normal (left) and polarized light (right with black background). (**A**,**B**) Hearts in StD-Doxo-treated mice increase the degree of fibrosis. Differently, the hearts of animals fed only EAAs show finer collagen fibers (**C**) than those observed in StD-fed animals. Doxo treatment did not increase the fibrosis (**D**). Scale bar 20 µm.

**Figure 2 nutrients-15-02287-f002:**
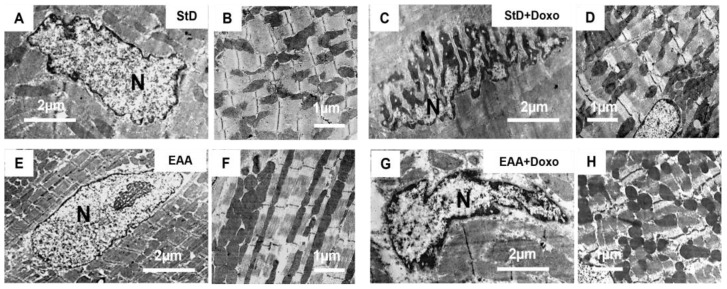
Representative TEM pictures of cardiomyocytes according to diet and Doxo treatment. Nucleus (**A**) and cytoplasm (**B**) of cardiomyocytes in mice fed with StD alone. After Doxo treatment we frequently observed irregular nuclei with bits-like signs. Sometimes these nuclei appeared very fragmented, with thicker condensed chromatin near the inner surface of the nuclear envelope (**C**). Furthermore, we observed mitochondria with irregular shapes and sizes and impaired distribution (**D**). Mice fed EAAs alone show a regular shape of the nucleus (**E**), and inside the cytoplasm, mitochondria are regularly oriented and distributed (**F**). After Doxo administration, the nuclear morphology appears to be like the control group, without bits-like signs and/or fragmentation, but with mild evidence of condensed chromatin (**G**). The mitochondria morphology and distribution (**H**) improves almost as much as with the EAAs diet. N = nucleus. Scale bar 2 μm (**A**,**C**,**E**,**G**) or 1 μm (**B**,**D**,**F**,**H**).

**Figure 3 nutrients-15-02287-f003:**
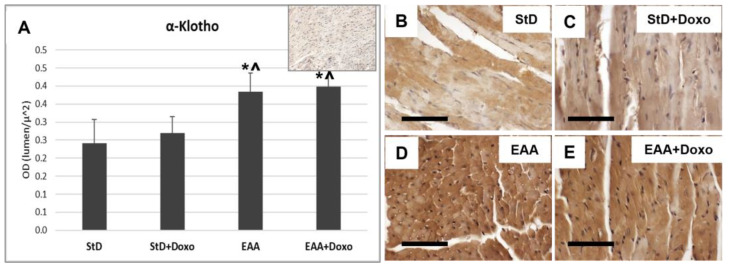
Klotho immunostaining intensity is expressed as optical density (**A**). In (**A**), the upper right image represents tissue incubated without primary antibody (blank control). Basal levels of klotho were faint in StD-fed mice (**B**), but increased strongly in EAA-fed mice (**D**). After Doxo administration, in both diets, the levels were comparable with that basal (**C**,**E**). ANOVA F = 22.09 *p* < 0.000. Scale bar 50 μm. *p* < 0.05 vs. * StD, ^ StD + Doxo.

**Figure 4 nutrients-15-02287-f004:**
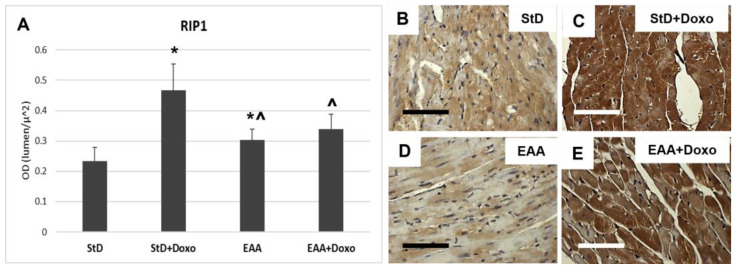
RIP1 immunostaining intensity is expressed as optical density (**A**). (**B**,**C**) RIP1 immunostaining before (**B**) and after (**C**) Doxo treatment in StD-fed mice. Doxo increases strongly the RIP1 staining. (**D**,**E**) Immunostaining staining before (**D**) and after (**E**) Doxo treatment in EAA-fed mice. The staining intensity increases in a non-homogeneous manner after Doxo. Indeed, many cells show faint staining. So, the staining intensity before and after treatment was non-significant. ANOVA F = 35.40 *p* < 0.000. Scale bar 50 μm. *p* < 0.05 vs. * StD, ^ StD + Doxo.

**Figure 5 nutrients-15-02287-f005:**
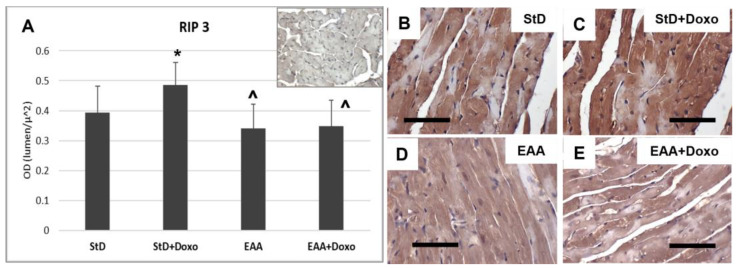
RIP3 immunostaining intensity is expressed as optical density (**A**). (**A**) The upper right image represents tissue incubated without primary antibodies (blank control). (**B**,**C**) RIP3 immunostaining before (**B**) and after (**C**) Doxo treatment in StD-fed mice shows a strong increase after chemotherapy (**C**). In the hearts of EAA-fed mice, the levels of RIP3 did not change before (**D**) or after chemotherapy (**E**). ANOVA F = 6.28 *p* < 0.002. Scale bar 50 μm. *p* < 0.05 vs. * StD, ^ StD + Doxo.

**Figure 6 nutrients-15-02287-f006:**
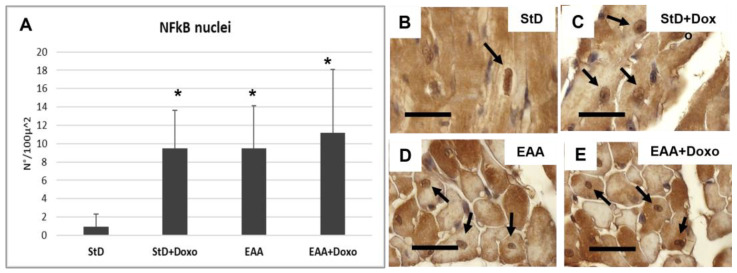
NFkB immunostaining intensity is expressed as a number of stained nuclei/100 μ^2^ (**A**). Stained nuclei increase strongly in StD-fed animals after Doxo administration. The EAA diet favors the increase of nuclear NFkB which does not change after chemotherapy. (**B**–**E**) Representative pictures of stained nuclei (arrows) according to diets and chemotherapy. ANOVA F = 9.71 *p* < 0.000. Scale bar 10 μm. *p* < 0.05 vs. * StD.

**Figure 7 nutrients-15-02287-f007:**
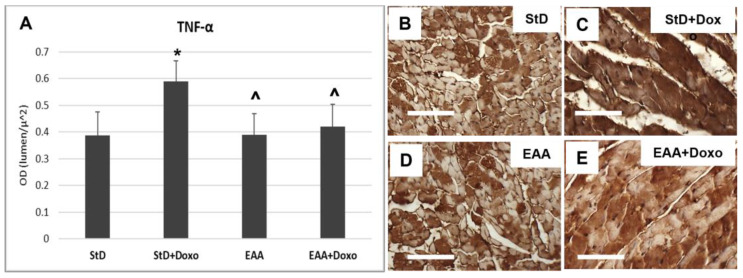
TNFα immunostaining intensity is expressed as optical density (**A**). In StD-fed animals, the immunoreaction increases strongly after chemotherapy (**B**,**C**). On the opposite side, the levels of TNFα do change before and after treatment in EAA-fed mice (**D**,**E**). ANOVA F = 14.84 *p* < 0.000. Scale bar 50 μm. *p* < 0.05 vs. * StD, ^ StD + Doxo.

**Figure 8 nutrients-15-02287-f008:**
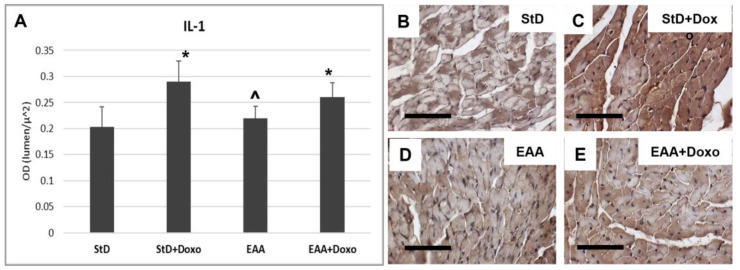
IL1 immunostaining intensity is expressed as optical density (**A**). In StD-fed animals, IL1 staining increases strongly after chemotherapy (**B**,**C**). On the opposite side, its level does change in EAA-fed mice (**D**,**E**). ANOVA F = 20.05 *p* < 0.000. Scale bar 50 μm. *p* < 0.05 vs. * StD, ^ StD + Doxo.

**Figure 9 nutrients-15-02287-f009:**
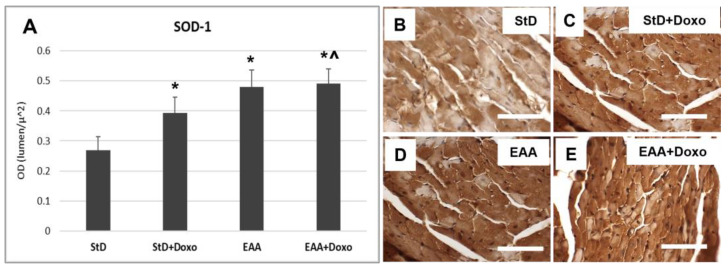
SOD1 immunostaining intensity is expressed as optical density (**A**). In StD-fed animals, SOD1 staining increases after chemotherapy (**B**,**C**). The EAA diet favors the early increase in SOD (**D**), which remains constant even after treatment with chemotherapy (**E**). ANOVA F = 44.19 *p* < 0.000. Scale bar 50 μm. *p* < 0.05 vs. * StD, ^ StD + Doxo.

**Figure 10 nutrients-15-02287-f010:**
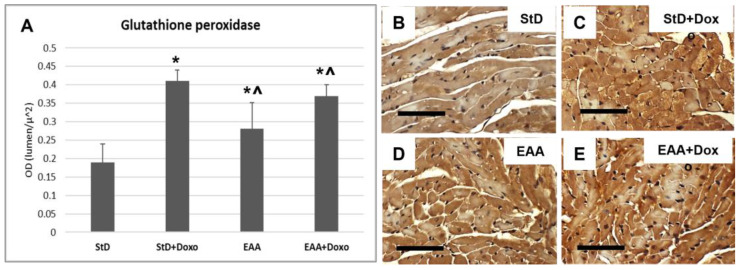
Glutathione peroxidase (Glu-px) immunostaining intensity is expressed as optical density (**A**). In StD-fed animals, Glu-px staining increases after chemotherapy (**B**,**C**). The EAA diet favors the early moderate increase in Glu-px (**D**), which increases much more after treatment with Doxo (**E**). ANOVA F = 41.85 *p* < 0.000. Scale bar 50 μm. *p* < 0.05 vs. * StD, ^ StD + Doxo.

**Figure 11 nutrients-15-02287-f011:**
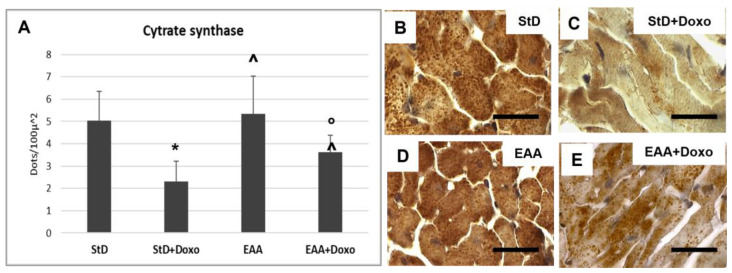
Citrate synthase (Cit-syn) immunostaining is expressed as number of stained dots/100 μm^2^ of cytoplasm (**A**). In StD-fed animals, Cit-syn staining decreases strongly after chemotherapy (**B**,**C**). The EAA diet favors the maintenance of Cit-syn level even after Doxo treatment (**D**,**E**). ANOVA F = 20.15 *p* < 0.000. Scale bar 15 μm. *p* < 0.05 vs. * StD, ° EAA, ^ StD + Doxo.

**Figure 12 nutrients-15-02287-f012:**
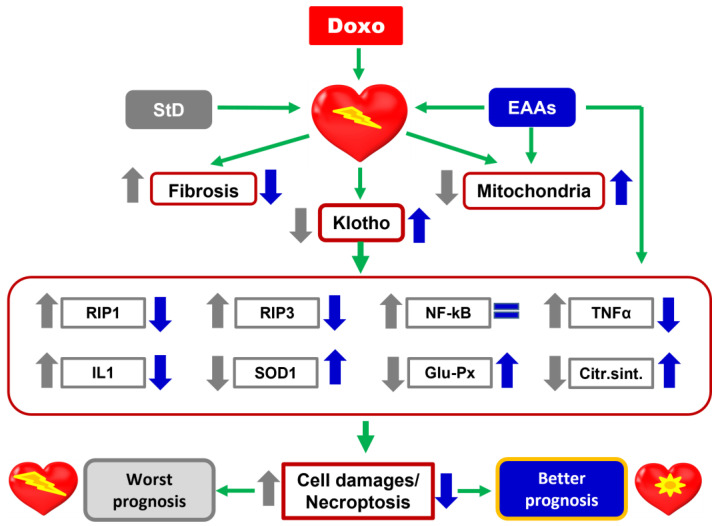
Schematic representation of the effects of StD and a special diet containing a mixture of all EAAs in the stoichiometric ratio on cardiac damages induced by Doxo. EAAs play a fundamental role in favoring the expression of Klotho and limiting the oxidative and inflammatory stress, and death of cardiomyocytes after treatment with chemotherapy. Therefore, an excess of EAAs in the diet has a protective action against the heart damages induced by Doxo.

**Table 1 nutrients-15-02287-t001:** Composition (qualitive and quantitative) of macronutrients in pellets fed to the two groups of mice. * Nitrogen (%) from free AAs only. ° Nitrogen (%) from vegetable and animal proteins and added AA. StD = standard diet; EAAs = essential AAs rich diet; N = nitrogen; bcaa = branched chain AAs. The bold line represents the limit between EAAs (upside) and non-EAAs (beneath). L-Cystine was included to match Sulphur AAs’ needs while minimizing methionine content.

	StD	EAAs
KCal/Kg	3952	3995
Carbohydrates (%)	54.61	61.76
Lipids (%)	7.5	6.12
Nitrogen (%)	21.8 °	20 *
Proteins: % of total N content	95.93	--
Free AA: % of total N content	4.07	100
EAA/NEAA (% in grams)	-	86/14
Free AA composition (%)		
l-Leucine (bcaa)	--	13.53
l-Isoleucine (bcaa)	--	9.65
l-Valine (bcaa)	--	9.65
l-Lysine	0.97	11.6
l-Threonine	--	8.7
l-Histidine	--	11.6
l-Phenylalanine	--	7.73
l-Methionine	0.45	4.35
l-Tyrosine	--	5.80
l-Triptophan	0.28	3.38
l-Cystine	0.39	8.20
l-Cysteine	--	--
l-Alanine	--	--
l-Glycine	0.88	--
l-Arginine	1.1	--
l-Proline	--	--
l-Glutamine	--	--
l-Serine	--	2.42
l-Glutamic Acid	--	--
l-Asparagine	--	--
l-Aspartic Acid	--	--
Ornithine-αKG	--	2.42
*N*-acetyl-cysteine	--	0.97

**Table 2 nutrients-15-02287-t002:** Changes in body weight (bw) and heart weight (hw) at the end of treatments. No statistical difference was observed between the two diets after chemotherapy treatment. ANOVA and Bonferroni T-test.

	Body Weight (g)	Heart Weight (g)	hw/bw (%)
StD	29.5 ± 1.41	0.2 ± 0.06	0.67 ± 0.17
StD + Doxo	29.29 ± 2.29	0.17 ± 0.02	0.60 ± 0.08
EAAs	28.5 ± 2	0.18 ± 0.01	0.63 ± 0.04
EAA + Doxo	28.02 ± 2.4	0.17 ± 0.02	0.61 ± 0.08
*F*	0.5	1.02	0.5
*p*	0.687	0.388	0.690

**Table 3 nutrients-15-02287-t003:** Morphometric data of mitochondria in diets before and after Doxo administration. Smit = surface of mitochondria. Scyt = total surface of cytoplasm. Nmit/100 μm^2^ = number of mitochondria in 100 μm^2^ of cytoplasm alias mitochondria density. ANOVA and Bonferroni *t*-test: *p* < 0.01 vs. * StD, ° EAAs, ^ StD + Doxo.

	Smit/Scyt (%)	Nmit/100 μ^2^	Smit Mean (μ^2^)
StD	19.84 ± 2.37	32.06 ± 3.49	0.621 ± 0.06
StD + Doxo	13.7 ± 1.57 *	23.01 ± 3.55 *	0.603 ± 0.09
EAAs	19.75 ± 2 ^	34.96 ± 7.3 ^	0.58 ± 0.1
EAA + Doxo	17.81 ± 2.14 * ^	29.84 ± 4.57 ° ^	0.612 ± 0.13
*F*	31.72	16.85	0.48
*p*	0.000	0.000	0.695

## Data Availability

All data generated or analyzed during this study are included in this published article.

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
