# Peer review of "Essential Amino Acids-Rich Diet Increases Cardiomyocytes Protection in Doxorubicin-Treated Mice"

_nutrients, 2023, doi:10.3390/nu15102287_

Round 1

Reviewer 1 Report

In the present study, the authors have aimed to portray the significance of EAA rich diet in preventing doxorubicin induced cardiomyocyte damage. Howeever, the study has some serious flaws:

In the introduction, there is no clear explanation for rationale of study for using EAA

There is no text explaining the connection between EAA and Klotho and EAA were abruptly introduced.

English grammar need to be thoroughly checked (example, pg 3, line 114.. “To this end…” needs to be rewritten)

In the materials and methods section, 2.1 (animals and treatments), the authors mentioned four group, however they just defined two groups which were further divided into two. This text should be rewritten more clearly.

As the authors acknowledged, IHC results can be misleading and can be easily manipulated, hence cannot be relied only on IHC for confirmation of results.

Authors are advised to do mRNA expressions for IL’s, NFkB, TNFa etc to have accurate data.

Further, protein expressions like WB, SOD activity assays, ROS activity studies etc. should be done to establish the results.

English grammar need to be thoroughly checked 

Author Response

We wish to thank the reviewer for the constructive criticism and the insightful suggestions. Accordingly, the revised version of our manuscript was amended to incorporate the reviewer’s recommendations, whereby improving the overall soundness of our study. Ad hoc answers were given below to every point raised by the reviewer.

1) In the introduction, there is no clear explanation for rationale of study for using EAA.

The rationale lies in the fact that enzymes (such as those we assessed) are generally proteins, which are composed of AA. In other terms, the supplementation of EAA can lead to biochemical and functional changes, which concept was now emphasized in the revised version of our manuscript (line 102 - 104).

2 - There is no text explaining the connection between EAA and Klotho and EAA were abruptly introduced.

Klotho is an enzyme mainly produced by the kidney, but also by other organs, including the heart. Although the role of klotho was extensively investigated in many conditions and different organs, to our knowledge, there is no study thus far showing a direct link between klotho’s expression levels and oral supplementation of AA. Indeed, as EAAs are required to allow unrestricted protein synthesis, we postulated that  a diet rich in EAA can also affect the synthesis of klotho, as well as a number of other proteins, with attendant functional consequences.

3) English grammar need to be thoroughly checked (example, pg 3, line 114.. “To this end…” needs to be rewritten).

Thank you for the kind suggestion. Such sentence was now rewritten.

4) In the materials and methods section, 2.1 (animals and treatments), the authors mentioned four group, however they just defined two groups which were further divided into two. This text should be rewritten more clearly.

Once again, thank you for the kind suggestion. The misleading sentence was amended accordingly.

5) As the authors acknowledged, IHC results can be misleading and can be easily manipulated, hence cannot be relied only on IHC for confirmation of results. Authors are advised to do mRNA expressions for IL’s, NFkB, TNFa etc to have accurate data. Further, protein expressions like WB, SOD activity assays, ROS activity studies etc. should be done to establish the results.

We are grateful to the referee for the observation. However, as indicated in §4.2 Limits of the study, we specified that ours is a histopathology work carried out by IHC and ultrastructural morphology approach. Moreover, we employed IHC optical density (OD) score, which is a new practical method for quantitative image analysis. More specifically, in order to reduce the visual bias intrinsic to manual estimation of antibody staining intensity, we resorted to an advanced digital image processing system (as reported in the Material and Methods session) which enables a more accurate and high volume IHC quantitative analysis by means of color deconvolution and computerized pixel profiling allowing automated scoring of the images. We do believe that our data, despite their intrinsic limitations, lay the groundwork for future more comprehensive studies, involving the production of functional and molecular data. 

6) Comments on the Quality of English Language. English grammar need to be thoroughly checked. 

An accurate linguistic revision was carried out by the two mother tongue contributors (TS and CCS).

Reviewer 2 Report

The manuscript written by Giovanni Corsetti et al., evaluated the protective role of essential amino acids against cardiotoxicity of doxorubicin. The paper is easy to follow and written very well.

Minor request:

-        Line 18: Please provide the complet name of ROS - reactive oxygen species (ROS).

Author Response

We thank the reviewer for the constructive criticism and the kind words of appreciation.

The acronym was now spelled out.

Round 2

Reviewer 1 Report

The authors have edited the manuscript according to the suggestions/comments, hence can be published now.